# Photoprotective Agents Obtained from Aromatic Plants Grown in Colombia: Total Phenolic Content, Antioxidant Activity, and Assessment of Cytotoxic Potential in Cancer Cell Lines of *Cymbopogon flexuosus* L. and *Tagetes lucida* Cav. Essential Oils

**DOI:** 10.3390/plants11131693

**Published:** 2022-06-27

**Authors:** Karina Caballero-Gallardo, Patricia Quintero-Rincón, Elena E. Stashenko, Jesus Olivero-Verbel

**Affiliations:** 1Environmental and Computational Chemistry Group, School of Pharmaceutical Sciences, Zaragocilla Campus, University of Cartagena, Cartagena 130014, Colombia; kcaballerog@unicartagena.edu.co (K.C.-G.); patriciaquintero@gmail.com (P.Q.-R.); 2Functional Toxicology Group, School of Pharmaceutical Sciences, Zaragocilla Campus, University of Cartagena, Cartagena 130014, Colombia; 3Center for Chromatography and Mass Spectrometry CROM-MASS, Research Center for Biomolecules CIBIMOL, School of Chemistry, Universidad Industrial de Santander, Bucaramanga 680006, Colombia; elena@tucan.uis.edu.co

**Keywords:** essential oils, sun protective factor, critical wavelength, cytotoxic potential, oxidative stress

## Abstract

Photoprotective agents obtained from plants provide benefits for the health of the skin. The present study aims to assess the total phenolic content (TPC) and in vitro UV-protective properties of twelve essential oils (EOs) from plants grown in Colombia and to evaluate the antioxidant and cytotoxic potential of two species identified as photoprotective potentials: *Cymbopogon flexuosus* and *Tagetes lucida*. The composition of EOs was studied by GC/MS. The cytotoxicity of both EOs was examined using an MTT assay, and an H_2_-DCFDA probe was employed to estimate the intracellular production of ROS in HepG2 and Calu-1 cells. Major constituents (≥10%) were neral, geranial, geranyl acetate in *C. flexuosus* and estragole in *T. lucida*. The TPC for *C. flexuosus* and *T. lucida* EOs were ≥10 mg GAE/g of byproduct. Both EOs showed photoprotective properties (SPF_in vitro_: 13–14), and long-wavelength UVA protection (λc > 370 nm). HepG2 and Calu-1 cells exposed to *C. flexuosus* exhibited antiproliferative activity (˂50%) at 125 µg/mL, while *T. lucida* was at 250 and 500 µg/mL. The IC_50_ values for *C. flexuosus* were 75 and 100 µg/mL in HepG2 and Calu-1 cells, respectively, whereas those for *T. lucida* were >250 µg/mL. These EOs achieved significant inhibitory effects (between 15.6 and 40.4%) against H_2_O_2_-induced oxidative stress. The results showed that EO compounds recognized as antioxidants could counteract the effects elicited by H_2_O_2_.

## 1. Introduction

Many aromatic species from the Asteraceae, Annonaceae, Lamiaceae, Poaceae, Rutaceae, Verbenaceae, and Zingiberaceae families grown in Colombia have important potential for the discovery of biologically active substances [1,2], because they produce endogenous antioxidants necessary for the neutralization of reactive oxygen species (ROS), including phenolics, flavonoids, anthocyanidins and anthocyanins, phenylpropanoids, stilbenes, terpenes and terpenoids, steroids, lignans, carotenoids, and essential oils (EOs) [3,4,5]. All these compounds play a significant role as defense mechanisms against infections and as a means of survival in an adverse environmental climate, e.g., in an environment rich in solar radiation [6]. Among the various plant-derived natural products, EOs are of great relevance because they can be used as active ingredients with health-improving properties when they are employed in the phytopharmaceutical and food industries [2,7,8].

The constituents of EOs have a wide range of molecular arrangements that include monoterpene and sesquiterpene hydrocarbons, monoterpenoids, sesquiterpenoids, phenylpropanoids, aliphatic compounds (alcohol, aldehydes), and some heterocyclic compounds. All these molecules are low in molecular weight, lipophilic, and highly volatile under normal conditions, and act as antiseptics, analgesics, anti-inflammatory agents, anti-proliferative agents, sedatives, anesthetics, and spasmolytics [9,10]. Most of the volatile constituents have a direct effect on molecular pathways, especially on the inflammatory signaling cascade, which involves the participation of nitric oxide, receptors such as opioids, and others [11,12,13].

Derivatives of anthranilates, benzophenones, camphor, cinnamates, dibenzoylmethane, *p*-aminobenzoates, and salicylates are organic compounds used as sunscreens in cosmetic formulations [14]. These molecules with aromatic structures have a carboxyl group that undergoes isomerization under the influence of energy absorbed from radiation [15]. However, there is a growing interest among consumers in cosmeceuticals that include natural products as part of a formulation. For this reason, considering the antioxidant capacity of phenolics, terpenes, and other molecules with the ability to reduce the adverse effects caused by solar radiation, natural products isolated from aromatic plants, including EOs, have been studied. In the literature, it has been reported that EOs act by modulating different cellular signaling pathways because they do not have a specific cellular target. Therefore, it is necessary to study the cellular antioxidant defense, which includes forkhead box O (Foxo), and nuclear factor erythroid-2 related factor 2 (Nrf-2), since Nrf-2 is essential in the protection of cells against oxidative damage and stimulates antioxidant capacity and phase II gene expression [16].

For decades, EOs have been used in degenerative skin disorders, and today, it is known that many of their protective properties are due to the presence of phenolic and terpene constituents [9,17]. The hydroxyl groups present in phenolic constituents exert antioxidant activity by inactivating free radicals by modulating cellular signaling transduction pathways, restoring the redox homeostasis by regulating cytoprotective responses caused by ROS and electrophiles, and preventing inflammation by enhancing the activities of antioxidant enzymes through the Nrf2 pathway [7,9,18,19]; thus, phenolic compounds reduce inflammation, oxidative stress, and DNA-damaging effects [20]. As photoprotective agents, these molecules have broad-spectrum UV absorption that includes regions between 290 nm and 400 nm. UV radiation is a dangerous environmental agent that causes molecular alterations and several harmful responses in skin cells [21,22,23,24,25]. This radiation range downregulates ROS elimination pathways, thereby promoting their production in mitochondria-mediated apoptosis [17]. UVA radiation (320 to 400 nm) can penetrate deeper into the dermis than UVB radiation (290 to 320 nm), but UVB radiation is more energetic than UVA radiation, which enables both radiation subtypes to adversely impact the epidermis [26]. UVB radiation induces the overproduction of ROS, which can react with lipids, nucleic acids, and proteins, causing oxidative stress and damage to cells [15,19]. Both UVA and UVB radiations produce DNA damage [22], mainly cyclobutene pyrimidine dimers, which are considered the main mutagenesis cause [23,27], causing skin cancer [28]. In keratinocyte cells, ROS generate apoptosis through the stimulation of the caspase-mediated signaling pathway [29]. However, phenolic compounds have been known to prevent keratinocytes apoptosis from UVB irradiation-induced photodamage [17]. One of the phenolic compounds most recognized for its photoprotective potential is the *trans*-resveratrol, a stilbenoid that inhibits UVB-induced inflammation and lipid peroxidation of the skin following topical application [15].

Currently, the cosmeceutical potential of EOs is being evaluated by the scientific community worldwide due to the need to find photoprotective agents that can be incorporated into sunscreen formulations to combat skin aging caused by UV radiation [30,31,32]. In this sense, the diversity of Colombian flora could be a rich source of these resources globally due to the privileged geographical position with sufficient natural resources and water, which makes it the second most biodiverse country after Brazil [2]. The addition of EOs to final products provides added value to cosmeceuticals due to the antioxidant properties of their chemical constituents; therefore, these components might be used as a complement to solar filters to reduce the oxidative stress produced by UV radiation [27], as well as by their anti-inflammatory, antimicrobial, antiaging, and anticancer properties [33]. The aim of the present study was to evaluate the in vitro UV-protective properties, and total phenolic content of twelve EOs, cytotoxic activities, and cellular-protective potential of *C. flexuosus* and *T. lucida* EOs against oxidative damage.

## 2. Results

### 2.1. Quantification of Total Phenolic Content in EOs

The results of the quantification of total phenolic content using gallic acid as a standard, evaluated through spectrophotometric analysis of twelve EOs distilled from Colombian plants, are shown in Table 1. *L. origanoides* (Carvacrol/thymol chemotype), *C. citratus*, *C. odorata*, and *L. origanoides* (Phellandrene chemotype) showed a phenolic content higher than 20 mg gallic acid equivalent (GAE)/g of byproduct, while *T. vulgaris*, *C. nardus*, *L. alba*, *C. sinensis*, *C. martinii*, *T. lucida*, *E. cardamomum*, and *C. flexuosus* showed a lower total phenolic content.

### 2.2. Assessment of UV Protective Properties

The SPF, λc, UVA/UVB ratio, the transmission of erythema (%), and transmission of pigmentation (%) determination of EOs are shown in Table 2. The SPF values ranged from 3.2 ± 0.0 to 14.7 ± 1.66 for the EOs evaluated, and the SPF results decreased in the following order: *T. lucida* > *C. flexuosus* > *L. origanoides* (Carvacrol/thymol) > *C. citratus* > *L. alba* > *C. odorata* > *C. martinii* > *T. vulgaris* > *E. cardamomum* > *C. nardus* > *L. origanoides* (Phellandrene) > *C. sinensis*. According to these results, all the EOs evaluated protect against UV radiation. The results found for λc indicated that the EOs from *C. flexuosus* and *T. lucida* EOs presented values higher than 370 nm. Regarding the UVA/UVB ratio, the results showed that 7/12 EOs were greater than 0.7, indicating that these EOs are potential photoprotective agents. In the case of transmission of erythema (%), *C. citratus*, *C. flexuosus*, *L. alba*, *L. origanoides* (Carvacrol/thymol chemotype), and *T. lucida* EOs with the lowest indices (1–6) were located within of the classification of extra protection, while for transmission of pigmentation (%), these same EOs except for *L. origanoides* (Phellandrene chemotype) were in the sunscreen range.

### 2.3. Determination of the Cytotoxic Potential of HepG2 and Calu-1 Cells in the MTT Viability Assay

The results obtained for the cytotoxic potential of EOs on HepG2 and Calu-1 cell lines are shown in Figure 1A,B and Table 3 and Appendix A. *C**. flexuosus* EO was more cytotoxic than *T. lucida* EO in HepG2 and Calu-1 cell lines. Both EOs displayed some proliferative activity at low concentrations, but this was not concentration dependent.

### 2.4. ROS Detection Assay of EOs in HepG2 and Calu-1 Cells

The results of the quantification of early intracellular ROS production by *C. flexuosus* and *T. lucida* EOs in HepG2 and Calu-1 cells are presented in Figure 2A,B, respectively. The detection of ROS levels was not significant when compared with the control. Hydrogen peroxide at a concentration of 200 µM (positive control) exhibited an elevated production of intracellular ROS during each recorded time (up to 120 min).

### 2.5. Assessment of Protective Effect against H_2_O_2_-Induced ROS

The protective effect against H_2_O_2_-induced ROS in HepG2, and Calu-1 cells exposed to each EO at a concentration of 15.6 µg/mL for 24 h is presented in Figure 3A,B, respectively. The results showed significant protective effects against H_2_O_2_-induced oxidative stress. HepG2 cells pretreated with *C. flexuosus* or *T. lucida* EOs showed a percentage of inhibition of ROS production (at 120 min) of 15.6% and 35.4%, respectively. Similarly, potential ROS inhibition was evaluated in Calu-1 cells pretreated with each EO, and a recovery percentage of 40.4% was observed with *C. flexuosus* EO, while with *T. lucida* EO, it was 35.8%.

## 3. Discussion

Currently, EOs are being explored as an active ingredient of cosmeceutical formulations to prevent and protect the skin from environmental damage [30]. The interest in these by-products is due to their antiseptic (i.e., bactericidal, virucidal, repellents, and fungicidal) [34,35,36,37], analgesic, anti-inflammatory, and antioxidative properties [38]. In this work, the cosmeceutical potential of twelve EOs obtained from plants grown in Colombia was determined by in vitro assessment of broad-spectrum UV-protection indices [39]. In addition, the total phenolic content was analyzed [25]. Based on the UV-protection indices, the two EOs with the best results were selected, and their cytotoxic activity was determined using cell lines of human hepatocellular carcinoma (HepG2) and lung squamous cell carcinoma (Calu-1), as well as their cytoprotective effects against oxidative stress induced by H_2_O_2_.

The genus *Cymbopogon* (Poaceae family) is used widely in traditional medicine [40,41]. Ethnopharmacology justifies its use as a pesticide or safe biocontrol agent to prevent stored product commodities from fungal pathogens [42], as an anti-ovicidal agent on *Aedes aegypti* (L.) eggs and influence larval development [43] and as an anti-inflammatory agent [11,44]. Moreover, *Cymbopogon* species have been used for the treatment of skin and respiratory infections, so they could be used as an ingredient in cosmetics, fragrances, and pharmaceutical formulations [44,45]. Scientific studies provide evidence of their use as an alternative to antiproliferative drugs [10,11,46]. *C. flexuosus* is a perennial aromatic grass widely used in tropical countries and has been studied for its medicinal value [47] and reported to have cytotoxic activities in cancer cell lines due to decreasing cell proliferation, altering mitochondrial membrane potential, increasing intracellular ROS, and initiating apoptosis [9,11,46,47,48]. Han and Parker reported the anti-inflammatory activity of *C. flexuosus* EO in human skin cells due to its ability to inhibit the levels of inflammatory biomarkers such as VCAM-1, IP-10, I-TAC, and MIG, as well as low levels of the tissue remodeling biomarkers collagen-I and III, EGFR, PAI-1, and the inhibition of the immunomodulatory biomarker M-CSF [49]. Furthermore, an antifungal activity study demonstrated that this EO has a higher activity than its pure components [11].

The antioxidant activity of *C. flexuosus* EO has been reported in the DPPH• radical scavenging assay using ascorbic acid as a standard antioxidant compound with an IC_50_ value of 43.67 µg/mL and inhibition (%) of 78 ± 1.11 at a concentration level of 150 µg/mL [44]. Microencapsulation of *C. flexuosus* EO is a solution to the low miscibility of oil in water and an unstable characteristic of EOs [50]. López et al. reported the antigenotoxic properties of geranial or citral, which are recommended for skin chemoprotection [51].

*Tagetes lucida* is commonly known as “pericón”, “hierba anis o hierba de Santa Maria”, “marigold”, and “winter tarragon” [52]. This species has been used as a herbal remedy for fever treatment, anxiety, irritability, depression, stomach disorders, rheumatism, tumors, asthma, and flu [53]. Pharmacological studies have reported antimicrobial, antileishmanial, anti-aggregate, hepatoprotective, antinociceptive, antispasmodic, and antihyperglycemic activities [52,53,54,55,56]. Olivero-Verbel et al. reported the antioxidant activity of *T. lucida* EO expressed in half-maximal effective concentration EC_50_ of 37.9 µg/mL using the thiobarbituric acid reactive species (TBARS) method [1]. Furthermore, anti-inflammatory activity has been attributed to both direct effects on the inflammatory signaling cascade and antioxidant activity [12]. Estragole (1-allyl-4-methoxybenzene) is the major constituent of *T. lucida* EO grown in Colombia. It is an aromatic compound methoxy derivative with potential antioxidant activity, which has restricted use due to possible carcinogenic effects in humans [57]. The literature shows a high concentration of estragole (95.7%) and some minor constituents, such as myrcene (1.6%) and *trans*-*β*-ocimene (1.0%), in *T. lucida* EO grown in Colombia [58]. According to a study carried out in Mexico, *T. lucida* OE contains geranyl acetate as the main constituent, followed by geraniol and *β*-caryophyllene [13]. Additionally, over 90% of estragole is reported in winter tarragons grown in Egypt [59]. Other authors indicate the existence of at least four chemotypes of this plant, where the major constituents are (*E*)-anethole (up to 74%), estragole (up to 97%), methyl eugenol (up to 80%), and nerolidol (approximately 40%) [59].

The antioxidant activity is a fundamental mechanism of the photoprotective activity of EOs and the extract of plants [25]. Skin damage by UV radiation depends on the generation of ROS, which includes peroxyl radicals (ROO•), hydroxyl radicals (OH•), and superoxide anions (O_2_•) [60]. The EOs produce medicinal effects through the combination of all active ingredients represented by monoterpenes, sesquiterpenes, phenolics, and their oxygenated derivatives [9,10,11,12,13]. The promising candidates for photoprotective agents identified in this investigation contain a mixture of terpenoids, phenolics, and oxygenated hydrocarbons, such as aliphatic aldehydes, alcohols, and esters, which have been identified as skin protective agents by other authors [13,44,45,51].

Skin photoaging is caused by ultraviolet radiation, which affects skin integrity by DNA damage, ROS, inflammation, immunosuppression, and activation of the neuroendocrine system, among others, causing skin disorders ranging from erythema, burns, dermatitis, cracks, hypo- and hyperpigmentation, immune suppression, oxidative stress, and finally skin cell death [61,62,63]. Parrado et al. reported that oxidative stress induced by ultraviolet radiation activates mitogen-activated protein kinases (MAPKs), nuclear factor kappa B (NF-κB), tumor necrosis factor-*α* (TNF-*α*), and proinflammatory cytokines (e.g., IL-1*β* and IL-6) and increases the transcription of matrix metalloproteinases (MMPs), causing dysregulation of extracellular matrix homeostasis and the degradation of collagen and elastin [64,65]. To counteract the effects of sunlight on human skin, the scientific community explores secondary metabolites derived from plants, including EOs.

In vitro assessment of UV-protection indices allowed the identification of two EOs as potential broad-spectrum sunscreens against long-wavelength UVA since their photoprotection indices are adjusted to the requirements of the literature [66,67]. This study identified the promising candidates *C. flexuosus* and *T. lucida* EOs, with SPF values of 13.4 and 14.7, λc values of 391.1 and 393.5 nm (long wavelength UVA-protection), UVA/UVB ratios of 0.7 ± 0.0 and 0.7 ± 0.0 (superior protector), the transmission of erythema (%) values of 1.4 ± 0.1 and 1.2 ± 0.0 (extra protection), and the transmission of pigmentation values of 20.3 ± 0.6 and 12.5 ± 0.1 (sunscreen), respectively. The SPF value obtained for both EOs allows them to be elevated to the category of active ingredients, since this index alone provides information on the preventive and protective properties that they would exert on skin exposed to solar radiation when is used as an active ingredient in cosmeceuticals. A quick review of the literature indicates that the SPF values obtained for *C. flexuosus* and *T. lucida* EOs from plants grown in Colombia are higher than the SPF values calculated for other plant species of the Asteraceae, Poaceae, and Geraniaceae families. In this regard, SPF values in EOs of 8.36 and 14.84 have been reported for *Calendula officinalis* [30,68], SPF of 6.282 for *Cymbopogon* spp. [39], an SPF of 2.299 for *Oncosiphon suffruticosum* [25] and an SPF of 6.45 for the *Pelargonium graveolens* EO [30]. The calculation of the photoprotection indices identified for the *C. citratus*, *L. alba*, and *L. origanoides* (Carvacrol/thymol chemotype) EOs indicated that they are promising candidates as broad-spectrum UVA and UVB sunscreens, with λc values of 360.9, 346.7, and 351.4 nm, respectively. In vitro assessment of photoprotection indices of *C. odorata*, *C. sinensis*, *C. martinii*, *C. nardus*, *E. cardamomum*, *L. origanoides* (Phellandrene chemotype), and *T. vulgaris* EOs resulted in values outside the range of UV sunscreens.

For MTT assays, *C. flexuosus* EO was evaluated at concentrations of 3.9–125 µg/mL, and *T. lucida* EO was evaluated at concentrations of 3.9–500 µg/mL. Proliferative activity was observed in Calu-1 cells treated with *C. flexuosus* EO at concentrations of 3.9–31.3 µg/mL, while no increase in cell viability was observed in HepG2 cells. However, *C. flexuosus* EO showed increased cytotoxicity in HepG2 cells, with an IC_50_ equal to 75 µg/mL, in comparison with Calu-1 cells, where the IC_50_ was 100 µg/mL. The *T. lucida* EO showed only proliferative activity in Calu-1 cells at concentrations of 3.91–125.0 µg/mL. Furthermore, this EO showed minor cytotoxicity in Calu-1 cells, with an IC_50_ of 381 µg/mL, in contrast with HepG2 cells, where the IC_50_ was 270 µg/mL. The ability of *C. flexuosus* EO to induce cytotoxic effects in cancer cell lines highlights its potential as an anticancer agent. Sharma et al. showed that the IC_50_ values of *C. flexuosus* EO ranged from 4.2–79 μg/mL, which depended upon the cell lines evaluated [48], while *T. lucida* EO (Estragole-chemotype) has been reported to have cytotoxicity values ranging between 80.2 and 156 μg/mL against murine macrophages and J774 cell lines [55]. The cytotoxic activity of *C. flexuosus* EO does suggest that further tests are needed before human exposure. The results obtained in MTT assays show that the cytotoxic potential for both EOs agrees with that reported by other authors [48,55,69].

ROS are considered to be signaling molecules that contribute to the natural aging process, inducing cellular differentiation and apoptosis [70]. Cancer cells have an elevated ROS level in comparison with normal cells, which permits the activation of pathways essential for both cellular transformation and tumorigenesis development [71]. In this study, HepG2 and Calu-1 human cancer cells were used for the assessment of early intracellular ROS using two concentrations of each EO (15.6 and 31.3 µg/mL). The results showed that *C. flexuosus* and *T. lucida* EOs at the concentrations tested do not induce significant intracellular ROS at 120 min. The literature reports that exposure of human cancer cells, e.g., HepG2 cells, to hydrogen peroxide significantly decreases cell viability by increasing ROS levels in a dose-dependent manner [72]. For this reason, both cell lines were used as a model of oxidative damage induced by H_2_O_2_ to determine the antioxidant potential or ROS-reducing potential of each EO.

The protective potential of the EOs against H_2_O_2_-induced reactive oxygen species was determined in vitro using cancer cell lines (HepG2 and Calu-1). The potential ROS reduction by *C. flexuosus* and *T. lucida* EOs was performed excluding cellular effects or mechanisms due to cytotoxicity through MTT viability assays in HepG2 and Calu-1 cells at 24 h. EOs have been widely studied as protective agents against cell damage induced by oxidative stress. Both the oxidative damage and inflammatory effects induced by hydrogen peroxide have been widely documented [73,74,75], and it is well known that oxidative stress caused by excessive ROS can lead to genotoxicity or DNA strand breaks [8]. *Salvia fruticosa* EO and its main constituents, including *α*-humulene, *α*-pinene, and *trans*-*β*-caryophyllene, have been highlighted as potential protective agents of cultured astrocytes against H_2_O_2_-induced cell death [76]. A study by Porres-Martinez et al. showed that PC12 cells pretreated with the Spanish sage (*Salvia lavandulifolia*) EO exhibited increased cell viability and decreases in both MDA levels and ROS production. Additionally, an increase in antioxidant enzymes (GSH:GSSG ratio) by activation of the transcription factor Nrf2 was observed [77]. Kang et al. showed protective effects of *Schisandrae semen* EO against H_2_O_2_-induced oxidative stress in C2Cl2 myoblast cells since this increases antioxidant capacity by activating the Nrf2/HO-1 pathway [78]. The results obtained in the present study highlight EOs as a rich source of protective agents that could be used against a wide variety of diseases related to oxidative stress, among them, atherosclerosis, periodontal, cardiovascular, and neurodegenerative diseases.

Finally, further studies should be carried out to assess the in vivo responses to EOs, both in terms of antioxidant defense and safety. This will expand the possibilities for development of novel, topical, cosmeceutical formulations with antioxidant and photoprotective potential.

## 4. Materials and Methods

### 4.1. Plant Material Collection, Extraction, and Characterization of EOs

The plant species used in this study were collected from experimental plots maintained at the Pilot Agroindustrial Complex of the National Center for Agroindustrialization of Aromatic and Medicinal Tropical Vegetal Species (CENIVAM, Bucaramanga, Colombia). The species were placed at the Colombian National Herbarium, with their respective vouchers (voucher number in parenthesis): *Cananga odorata* “ylang-ylang” (COL 531012); *Citrus sinensis* “orange” (CENIVAM-455); *Cymbopogon citratus* “lemon tea” COL 531013); *Cymbopogon flexuosus* “Lemon grass” (CENIVAM 462); *Cymbopogon martinii* “palmarrosa” (COL 587116); *Cymbopogon nardus* (COL 578357); *Elettaria cardamomum* “cardamom” (CENIVAM-450); *Lippia alba* “quick relief” (COL 480750); *Lippia origanoides*, phellandrene-rich chemotype (COL 519798); *Lippia origanoides* “wild oregano” carvacrol/thymol-rich chemotype (COL 512075); *Tagetes lucida* “winter tarragon, marigold” (COL 512074); and *Thymus vulgaris* “thyme¨ (COL 555843). The EOs were extracted by hydrodistillation (HD) or microwave-assisted hydrodistillation (MWHD). The EOs were analyzed by GC/MS and GC/FID as previously described [34,36,43,79,80] (Table 4).

### 4.2. Determination of Total Phenolic Content (TPC)

The determination of TPC of the EO samples was carried out using the Folin–Ciocalteu reagent method with modifications [81]. Briefly, 250 μL of essential oil (in methanol) was mixed with 2.5 mL of distilled water, followed by 125 μL of 1 N Folin–Ciocalteu reagent, and stirred for 5 min. Finally, 375 μL of 20% (*w*/*v*) Na_2_CO_3_ solution was added and kept in the dark for 2 h at room temperature. Absorbance was read at 760 nm using a Varioskan™ LUX Multimode Microplate Reader (Thermo Fisher Scientific, Inc., Waltham, MA, USA). Total phenolic contents were calculated using a gallic acid standard curve (A_760_ = 0.0046 [gallic acid] + 0.044, R^2^ = 0.9966), obtained using seven concentrations of gallic acid (0.003–0.200 mg/mL). Phenolic contents were expressed as milligrams of gallic acid equivalents per gram of byproduct (mg GAE/g of byproduct). Each sample was measured with three replications.

### 4.3. In Vitro Assessment of the UV Protective Properties of EOs

*C. odorata*, *C. sinensis*, *C. citratus*, *C. flexuosus*, *C. martinii*, *C. nardus*, *E. cardamomum*, L. alba, *L. origanoides* (Phellandrene chemotype), *L. origanoides* (Carvacrol/thymol chemotype), *T. lucida*, and T. vulgaris EOs were analyzed for sun protection factor (SPF), critical wavelength (λc), UVA/UVB ratio, the transmission of erythema (%) and transmission of pigmentation (%) by spectrophotometric analysis. The absorption of the twelve EO samples was evaluated between wavelengths of 290 and 400 nm using a Varioskan™ LUX Multimode Microplate Reader (Thermo Fisher Scientific, Inc., Waltham, MA, USA).

#### 4.3.1. Sun Protection Factor (SPF)

The SPF was determined from the values obtained of absorbance between wavelengths of 290 to 320 nm (UVB spectrum) with 5.0 cm of optical path and was applied to the Mansur equation [82].
(1)$$\mathrm{FPS}=\mathrm{CF}\;\times \sum_{290\;\mathrm{nm}}^{320\;\mathrm{nm}}\mathrm{EE}\;\left(\lambda \right)\times \;I\left(\lambda \right)\times \;\mathrm{Abs}\;\left(\lambda \right)\;$$
where CF is the correction factor (equal to 10) and EE(λ) x I(λ) are constants at each wavelength (Appendix A) obtained from the correlation between EE (erythemal effect of radiation of wavelength λ) and I (the solar intensity at wavelength λ) [83]. Finally, Abs is the absorbance of the solution at wavelength λ [33]. In this equation, the CF value was determined so that a standard sunscreen formulation containing 8% homosalate presented an SPF value of 4, determined by UV spectrometry [60].

#### 4.3.2. Critical Wavelength (λc)

The λc of the EOs was calculated using the following equation:(2)$$\int_{290\;\mathrm{nm}}^{\lambda c}A\;\left(\lambda \right)d\lambda =0.9\;\int_{290\;\mathrm{nm}}^{400\;\mathrm{nm}}A\left(\lambda \right)d\lambda \;$$
where A is the absorbance to the wavelength λ (nm). The area under the absorbance curve was considered to be 100%. λc is the wavelength at which 90% of the area under the absorbance curve is found considering the integral of the absorption spectrum from 290 to 400 nm with a 1.0 cm optical path [20]. For the FDA, a product is considered to have a broad spectrum of photoprotection when its minimum λc is 370 nm [84], i.e., the product must be able to absorb enough UV radiation at a longer wavelength (UVA) [66,67,85].

#### 4.3.3. UVA/UVB Ratio

The UVA/UVB ratio is the mean absorbance extinction in the spectral regions that includes both UVA (320 to 400 nm) and UVB (290 to 320 nm) [20]. The measurement of UVA/UVB ratios was determined according to the Boots Star Rating System. Briefly, the average UVA absorbance (320 to 400 nm) with 5.0 cm of optical path was related to the average UVB absorbance (290 to 320 nm) using the following equation:(3)$$\frac{\mathrm{UVA}}{\mathrm{UVB}}=\frac{\sum_{320\;\mathrm{nm}}^{400\;\mathrm{nm}}A\left(\lambda \right),\;d\left(\lambda \right)}{\sum_{290\;\mathrm{nm}}^{320\;\mathrm{nm}}A\left(\lambda \right),\;d\left(\lambda \right)\;}\;$$
where A(λ) is the effective absorbance related to the transmittance of the sunscreen. The star rating system indicates that the UVA/UVB ratio in 0.0 to <0.2 is too low for UVA protection (-), 0.2 to <0.4 is a moderate protector (*), 0.4 to <0.6 is a good protector (**), 0.6 to <0.8 is a superior protector (***), and 0.8 to ≥0.8 is a maximum protector (****) [20,84,86,87].

#### 4.3.4. Transmission of Erythema (%) and Transmission of Pigmentation (%)

The transmission of erythema (%) and transmission of pigmentation (%) was determined from the values obtained of absorbance between wavelengths of 292 to 372 nm with 5.0 cm of optical path. Both photoprotection indices were calculated from transmission (T), which is calculated by the formula A = 1/T = −log T and was applied to the following equations:(4)$$\mathrm{Transmission}\;\mathrm{of}\;\mathrm{erythema}\;\left(\%\right)=\frac{\mathrm{Ee}}{\sum \mathrm{Fe}}=\sum \frac{\left(T\;\times \;\mathrm{Fe}\right)}{\sum \mathrm{Fe}}\;$$
where Fe is the flux of erythema whose values are constant and correspond to wavelength between 292 and 338 nm. And
(5)$$\mathrm{Transmission}\;\mathrm{of}\;\mathrm{pigmentation}\;\left(\%\right)=\frac{\mathrm{Ep}}{\sum \mathrm{Fp}}=\sum \frac{\left(T\;\times \;\mathrm{Fp}\right)}{\sum \mathrm{Fp}}\;$$
where Fp is the flux of pigmentation, whose values are constant and correspond to wavelengths between 322 and 372 nm [61]. The constants used to calculate the erythema and pigmentation flux are presented in Appendix A [88], and the sunscreen category following the range of transmission UV (%) is shown in Appendix A.

### 4.4. Maintenance of Cell Lines

Human lung epidermoid carcinoma cells (Calu-1) and human hepatocellular carcinoma cells (HepG2) were obtained from the Cell Lines Service (CLS; Eppelheim, Germany), the American Type Tissue Culture Collection (ATCC, USA), and ScienCell (Carlsbad, CA, USA). Cell lines were maintained using protocols previously reported in the literature [89]. Briefly, Calu-1 and HepG2 cells were incubated at 37 °C and 5% CO_2_ in Eagle’s minimum essential medium (EMEM, Quality Biological, Gaithersburg, MD, USA) enriched with fetal bovine serum (FBS; Biowest, South America). To avoid the proliferation of pathogens, 1% penicillin/streptomycin (Gibco, Grand Island, NY, USA) was added.

### 4.5. Determination of the Cytotoxic Potential of HepG2, and Calu-1 Cells in the MTT Viability Assay

Cell viability was examined using the MTT assay as previously reported [90,91]. Briefly, a total of 2 × 10^4^ cells per well (200 μL of culture medium) were seeded in a 96-well plate and incubated at 37 °C in 5% CO_2_ for 24 h. Then, the culture media was discharged to expose the cells to eight oil concentrations (3.9–500 µg/mL) and incubated for 24 h at 37 °C and 5% CO_2_. Later, the medium with treatments was removed, and each well received 200 µL of medium with 50 µL of MTT solution (5 mg/mL). After an incubation period of 4 h at 37 °C, the MTT-containing media was discarded, and 200 µL of DMSO (Panreac Applichem^®^, Barcelona, Spain) was added to each well prior to reading the optical density at 570 nm using a Varioskan^TM^ LUX Multimode Microplate Reader (Thermo Fisher Scientific, Inc., Waltham, MA, USA). All experiments were performed in three independent experiments with four replicates for each treatment. Therefore, the percentages of cell viability were normalized and calculated with the following equation:(6)Cell viability (%)=(ODtest/ODcontrol)×100 
where OD is the optical density or absorbance of the test or control, respectively.

### 4.6. ROS Detection Assay of EOs in HepG2 and Calu-1 Cells

The antiaging effect of *C. flexuosus* and *T. lucida* EOs on early intracellular ROS production was monitored in HepG2 and Calu-1 cells by using a 2′,7′-dichlorodihydrofluorescein diacetate (H_2_-DCFDA) probe (Sigma-Aldrich, USA). DCFH_2_-DA is able to diffuse through the cell membrane and become enzymatically hydrolyzed by intracellular esterases to produce nonfluorescent DCFH_2_. The oxidation of DCFH_2_ by intracellular ROS, mainly H_2_O_2_, HO•, ROO•, NO•, and ONOO^−^, results in a highly fluorescent end product (DCF) that stains the cells. Hence, the intracellular ROS generation of cells can be investigated using DCFH_2_-DA as an indicator to detect and quantify intracellular reactive oxygen species [92,93]. The treatment concentrations were obtained by considering the cytotoxicity assays. The detection of ROS in HepG2 and Calu-1 cells treated in situ with different concentrations of both EOs, pretreated HepG2 and Calu-1 cells and then exposed to H_2_O_2_ (200 µM) was carried out as follows: 2 × 10^4^ cells per well were seeded in a 96-well black culture microplate (Greiner Bio-one, Frickenhausen, Germany) and cultured for 24 h at 37 °C and 5% CO_2_. For the in situ assay, cells were loaded with 20 μM H_2_-DCFDA in fresh medium for 20 min. Subsequently, H_2_-DCFDA was removed, and cells were washed with PBS and then exposed to *C. flexuosus* and *T. lucida* EOs (15.63 and 31.23 µg/mL) as an individual treatment. For the protection assay of H_2_O_2_-induced ROS, HepG2 and Calu-1 cells were exposed for 24 h to 15.63 µg/mL, both *C. flexuosus* and *T. lucida* EOs. The pretreatment was removed, and the cells were washed with PBS. Then, H_2_O_2_ (200 µM) was added to induce the generation of ROS. Increases in fluorescence were measured using a Varioskan^TM^ LUX Multimode Microplate Reader (Thermo Fisher Scientific, Inc., Waltham, MA, USA) at intervals up to 2 h excitation/emission wavelengths of 485/535 nm. The results are expressed as increase in fluorescence with respect to the control (untreated cells). For the protection assay, the results are expressed as increase in fluorescence with respect to a positive control (H_2_O_2_, 200 µM). Two independent experiments were performed with four replicates each.

### 4.7. Statistical Analysis

The results are presented as the mean ± standard error of the mean (X ± SEM). The cell viability and intracellular ROS are reported as the IC_50_ values, calculated using a nonlinear regression log of the EO concentrations versus the cell proliferation and fluorescence, respectively. Statistically significant differences in the concentrations when compared to the control and the variances were calculated with one-way ANOVA, with the application of Dunnett’s multiple comparison posttest, for both cell viability and intracellular ROS levels (*p* < 0.05). The confidence interval for all the analyses was 95%. The results were analyzed using GraphPad Prism 8.0.

## 5. Conclusions

The risk of increase in solar ultraviolet radiation on the Earth’s surface due to stratospheric ozone depletion [94], scientific curiosity, and industrial demand have been the driving forces behind studies carried out in plants to detect natural photoprotective compounds that allow the skin cells integrity to be preserved, preventing, or minimizing photoaging. Compounds present in EOs suggest that their radical scavenging action could have a practical application for treating human skin against oxidative damage as it undergoes environmental and chronological aging. The results obtained in the present study will broaden the scientific knowledge of the applications of EOs obtained from *C. flexuosus* and *T. lucida* in cosmeceutical formulations as topical options complementary to sunscreen development, which are focused on decreasing all damage from sun exposure, preserving the redox homeostasis, ensuring the integrity of the skin, and conferring pleasant sensory properties to the formulations.

## Figures and Tables

**Figure 1 plants-11-01693-f001:**
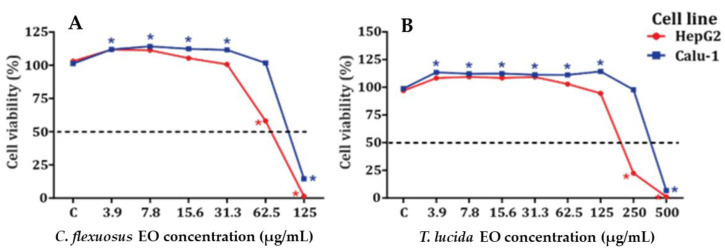
Cytotoxicity in HepG2 (**A**) and Calu-1 (**B**) cell lines exposed to *C. flexuosus* and *T. lucida* EOs for 24 h. * Significant difference in viability when compared to control group (C) (*p* < 0.05). Data are the mean ± SEM (*n* = 3).

**Figure 2 plants-11-01693-f002:**
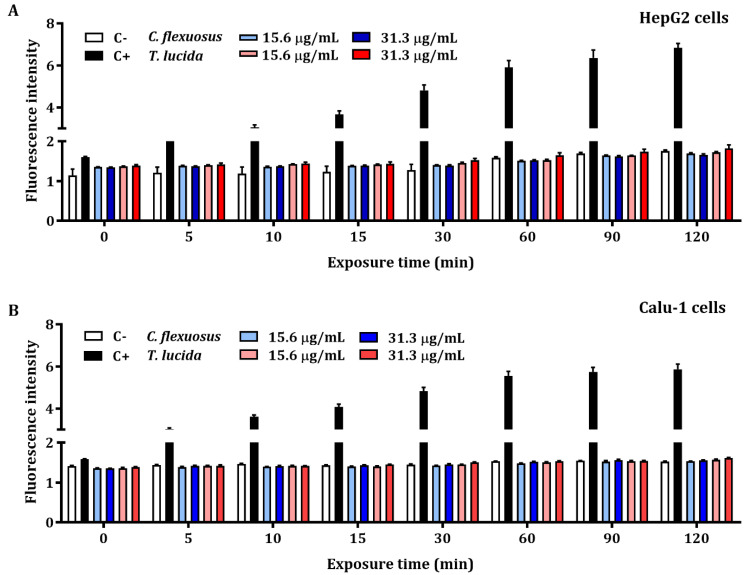
Early intracellular ROS assessment of concentrations based on MTT assays of *C. flexuosus* and *T. lucida* EOs on HepG2 (**A**) and Calu-1 (**B**) human cell lines. Culture medium and hydrogen peroxide (200 μM) served as negative (C-) and positive controls (C+), respectively. Significant differences were not observed when compared to the negative control group.

**Figure 3 plants-11-01693-f003:**
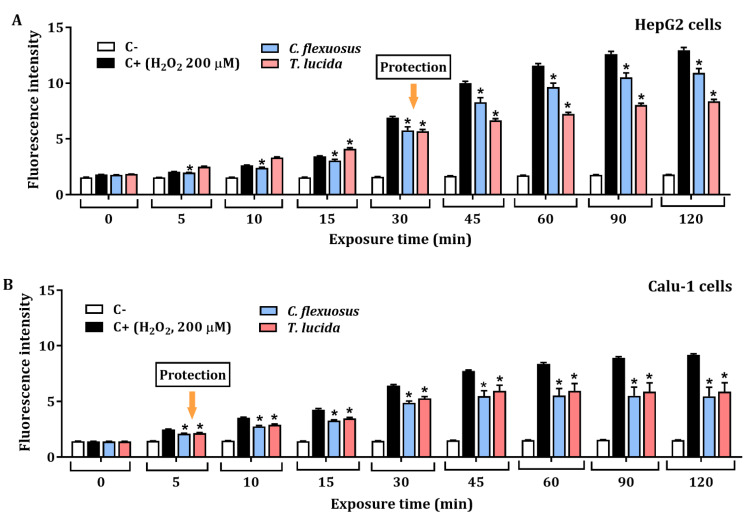
Protective effect at 15.6 µg/mL of *C. flexuosus* and *T. lucida* EOs against H_2_O_2_-induced ROS in HepG2 (**A**) and Calu-1 (**B**) cell lines. Cells were pretreated with each EO for 24 h. Culture medium and H_2_O_2_ (200 μM) served as negative (C-), and positive control (C+), respectively. * Significant difference when compared to the positive control (*p* < 0.05).

**Table 1 plants-11-01693-t001:** Quantification of total phenolic content in essential oils.

Essential Oil	Mean ± SEM
	Phenolic Content (mg GAE/g of ByProduct)
*Cananga odorata*	24.4 ± 0.4
*Citrus sinensis*	13.8 ± 0.9
*Cymbopogon citratus*	26.6 ± 0.3
*C. flexuosus*	10.6 ± 0.1
*C. martinii*	13.6 ± 0.1
*C. nardus*	18.6 ± 0.2
*Elettaria cardamomum*	12.7 ± 0.1
*Lippia alba*	16.4 ± 0.1
*L. origanoides* (Phellandrene)	21.9 ± 0.1
*L. origanoides* (Carvacrol/thymol)	26.9 ± 0.1
*Tagetes lucida*	12.7 ± 0.1
*Thymus vulgaris*	19.0 ± 0.1

**Table 2 plants-11-01693-t002:** In vitro assessment of UV protective properties of essential oils.

Essential Oils	In vitro Measurements of Sunscreen Protection (Mean ± SEM), *n* = 3				
	SPF ^a^	λc	UVA/UVB Ratio	Transmission of Erythema (%)	Transmission of Pigmentation (%)
*C. odorata*	8.7 ± 0.0	ND	0.4 ± 0.0	4.4 ± 0.0	43.3 ± 0.2
*C. sinensis*	3.2 ± 0.0	ND	1.1 ± 0.1	17.0 ± 0.1	71.9 ± 0.4
*C. citratus*	10.0 ± 0.2	360.9 ± 1.8	1.2 ± 0.0	3.5 ± 0.2	23.5 ± 0.7
*C. flexuosus*	13.4 ± 0.3	391.1 ± 1.9	0.7 ± 0.0	1.4 ± 0.1	20.3 ± 0.6
*C. martini*	5.2 ± 0.0	ND	0.6 ± 0.0	10.0 ± 0.1	55.9 ± 0.2
*C. nardus*	4.5 ± 0.0	ND	0.7 ± 0.0	12.0 ± 0.2	59.8 ± 0.3
*E. cardamomum*	4.7 ± 0.0	ND	0.7 ± 0.0	11.3 ± 0.2	57.5 ± 0.5
*L. alba*	9.6 ± 0.1	346.7 ± 1.5	1.0 ± 0.0	3.4 ± 0.0	29.8 ± 0.0
*L. origanoides* (Phellandrene)	3.6 ± 0.0	ND	0.3 ± 0.0	15.5 ± 0.4	70.4 ± 0.9
*L. origanoides* (Carvacrol/thymol)	11.7 ± 0.2	351.4 ± 0.0	0.1 ± 0.0	1.9 ± 0.0	63.9 ± 0.9
*T. lucida*	14.7 ± 0.0	393.5 ± 0.2	0.7 ± 0.0	1.2 ± 0.0	12.5 ± 0.1
*T. vulgaris*	4.8 ± 0.0	ND	0.5 ± 0.0	11.1 ± 0.0	58.1 ± 0.1
Trolox ^b^	3.6 ± 0.0	371.6 ± 0.5	0.1 ± 0.0	38.3 ± 0.1	71.3 ± 0.2
Vanillin ^b^	56.4 ± 0.9	375.1 ± 0.0	0.6 ± 0.0	0.0 ± 0.0	5.5 ± 0.1
Gallic acid ^b^	34.7 ± 0.1	348.2 ± 0.4	0.0 ± 0.0	0.0 ± 0.0	45.3 ± 0.5
Body lotion ^c^	36.3 ± 0.1	392.6 ± 0.3	0.4 ± 0.1	0.02 ± 0.2	11.7 ± 0.1
Sun protector ^c^	54.3 ± 0.2	393.1 ± 0.1	2.7 ± 0.2	0.0 ± 0.0	0.0 ± 0.0

^a^ Values at a concentration of 0.75 mg/mL. ^b^ Standard reagent (0.75 mg/mL). ^c^ Commercial product (sunscreen). ND, not determined (indices outside the valuated range).

**Table 3 plants-11-01693-t003:** Cytotoxic potential of *C. flexuosus* and *T. lucida* EOs on HepG2 and Calu-1 cells.

Essential Oil	Cytotoxic Potential Derived from MTT Viability Assay (24 h)					
	HepG2			Calu-1		
	IC_50_	R^2^	*p*-Value	IC_50_	R^2^	*p*-Value
*Cymbopogon flexuosus*	75	0.986	<0.001	100	0.865	0.007
*Tagetes lucida*	270	0.912	<0.001	381	0.856	0.001

IC_50_: Half-maximal inhibitory concentration (µg/mL).

**Table 4 plants-11-01693-t004:** List of the specimens studied. For each plant’s essential oils, the major constituents (>10%) are shown.

Plant	Family	Major Constituents (%)	Reference
*C. odorata*	Annonaceae	Ethyl benzoate (18.2), linalool (14.0), benzyl benzoate (12.3), methyl benzoate (10.0)	[43]
*C. sinensis*	Rutaceae	Limonene (71.3)	[43]
*C. citratus*	Poaceae	Geranial (34.4), neral (28.4), geraniol (11.5)	[43]
*C. flexuosus*	Poaceae	Neral (28.2), geranial (28.2), geranyl acetate (10.0)	[43]
*C. martinii*	Poaceae	Geraniol (83.9)	[80]
*C. nardus*	Poaceae	Citronellal (25.3), citronellol (17.9), geraniol (11.6)	[34]
*E. cardamomum*	Zingiberaceae	1,8-Cineole (30.9), terpinyl acetate (26.4)	[36]
*L. alba*	Verbenaceae	Carvone (38.3), limonene (31.8), bicyclo- sesquiphellandrene (11.4)	[43]
*L. origanoides* (Phellandrene chemotype)	Verbenaceae	Limonene (15.0), *p*-cymene (14.6), *α*-phellandrene (10.3)	[80]
*L. origanoides* (Carvacrol/thymol chemotype)	Verbenaceae	Carvacrol (50.6), thymol (11.5)	[43]
*T. lucida*	Asteraceae	Estragole (95.7)	[43]
*T. vulgaris*	Lamiaceae	Thymol (42.0), *p*-cymene (26.4)	[80]

## Data Availability

All data included in the main text.

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
