# Peer review of "Photoprotective Agents Obtained from Aromatic Plants Grown in Colombia: Total Phenolic Content, Antioxidant Activity, and Assessment of Cytotoxic Potential in Cancer Cell Lines of Cymbopogon flexuosus L. and Tagetes lucida Cav. Essential Oils"

_plants, 2022, doi:10.3390/plants11131693_

Round 1

Reviewer 1 Report

This manuscript contains sufficient novelty to be accepted for publication, but still minor modifications and suggestions are recommended to improve the quality.

  • All minor remarks are highlighted in the manuscript.
  • Authors should avoid the lumping of references in the paper, but each should be discussed.
  • A list of abbreviations should be given in the manuscript.

Author Response

Page 3, Line 102. Verb correction “provides

Answer:

The verb is used in the right form (singular).  Thus, no change was executed.

Page 3, Line 117

Give the full name of the abbreviation (GAE), because it is first mentioned in the paper

Answer:

The full name of the abbreviation “gallic acid equivalent (GAE)” was added.

Page 3, Table 1

Delete “Identified compounds”

Answer:

The words “Identified compounds” were deleted

Page 3, Table 1

Replace 'compounds' by 'content'

Answer:

The word “content” was added

Reviewer 2 Report

Please see the Manuscript

Author Response

Thanks to the Reviewer.

No corrections were suggested.

Reviewer 3 Report

The authors studied the total phenolic content (TPC) and in vitro UV-protective properties of twelve essential oils (EOs) from plants grown in Colombia and to evaluate the antioxidant and cytotoxic potential of two species identified as photoprotective potentials: Cymbopogon flexuosus and Tagetes lucida . The manuscript is quite well done and is interesting. The MS can be accepted for publication after the following minor modifications: In conclusion, the authors should place more emphasis on the implementation potential of the results. Above all, it would be appropriate to propose further directions of research (works) that will lead to implementation, ie to the use of their proposed EOs in production (toxicological tests, sensitivity tests, stability tests, etc.).

Author Response

Answer:

Thanks for the suggestion.

The following paragraphs were added to the final of the Discussion Section.

The results obtained in the present study highlight EOs as a rich source of protective agents that could be used against a wide variety of diseases related to oxidative stress, among them, atherosclerosis, periodontal, cardiovascular, and neurodegenerative diseases.

Finally, further studies should be carried out to assess the in vivo responses to EOs, both in terms of antioxidant defense and safety. This will expand the possibilities for development of novel, topical, cosmeceutical formulations with antioxidant and photoprotective potential.